# Thin-Film Engineering of Mechanical Fragmentation Properties of Atomic-Layer-Deposited Metal Oxides

**DOI:** 10.3390/nano10030558

**Published:** 2020-03-19

**Authors:** Mikko Ruoho, Janne-Petteri Niemelä, Carlos Guerra-Nunez, Natalia Tarasiuk, Georgina Robertson, Aidan A. Taylor, Xavier Maeder, Czeslaw Kapusta, Johann Michler, Ivo Utke

**Affiliations:** 1Empa–Swiss Federal Laboratories for Materials Science and Technology, Laboratory for Mechanics of Materials and Nanostructures, Feuerwerkerstrasse 39, CH-3602 Thun, Switzerland; mruoho@iki.fi (M.R.); janne-petteri.niemelae@empa.ch (J.-P.N.); Carlos.Guerra-Nunez@empa.ch (C.G.-N.); tarasiuknatalia@gmail.com (N.T.); georginar018@gmail.com (G.R.); xavier.maeder@empa.ch (X.M.); Johann.Michler@empa.ch (J.M.); 2Materials Department, University of California, Santa Barbara, CA 93106, USA; aidantaylor@ucsb.edu; 3AGH University of Science and Technology Krakow, Faculty of Physics and Applied Computer Science, Al.Mickiewicza 30, 30-059 Kraków, Poland; kapusta@agh.edu.pl

**Keywords:** atomic layer deposition, crack onset strain, fracture mechanics, interfacial shear strain, residual strain, saturated crack density, Al_2_O_3_-Y_2_O_3_-ZrO_2_-TiO_2_-ZnO, nanolaminate, uniaxial tensile strain

## Abstract

Mechanical fracture properties were studied for the common atomic-layer-deposited Al_2_O_3_, ZnO, TiO_2_, ZrO_2_, and Y_2_O_3_ thin films, and selected multilayer combinations via uniaxial tensile testing and Weibull statistics. The crack onset strains and interfacial shear strains were studied, and for crack onset strain, TiO_2_/Al_2_O_3_ and ZrO_2_/Al_2_O_3_ bilayer films exhibited the highest values. The films adhered well to the polyimide carrier substrates, as delamination of the films was not observed. For Al_2_O_3_ films, higher deposition temperatures resulted in higher crack onset strain and cohesive strain values, which was explained by the temperature dependence of the residual strain. Doping Y_2_O_3_ with Al or nanolaminating it with Al_2_O_3_ enabled control over the crystal size of Y_2_O_3_, and provided us with means for improving the mechanical properties of the Y_2_O_3_ films. Tensile fracture toughness and fracture energy are reported for Al_2_O_3_ films grown at 135 °C, 155 °C, and 220 °C. We present thin-film engineering via multilayering and residual-strain control in order to tailor the mechanical properties of thin-film systems for applications requiring mechanical stretchability and flexibility.

## 1. Introduction

Atomic layer deposition (ALD) is a thin-film deposition technique for the fabrication of conformal coatings with sub-nanometer-scale thickness control. A wide range of industrially relevant metal oxides and other materials can be deposited with ALD for many applications in the fields of microelectronics, energy generation and storage, and protective coatings. Much of the research on ALD has focused on the surface chemistry of the ALD processes and on the growth and film characteristics on a wide variety of surfaces. Regardless of the strong growth of the ALD field, the mechanical properties of the ALD-fabricated thin films have received relatively little attention.

Mechanical robustness of coating materials is desired for applications where the materials are subjected to straining. This is the case for coatings on flexible substrates, or coatings on materials that exhibit significant volumetric changes (e.g., owing to large changes in temperature). Typically, the mechanical robustness alone is not enough for high-performance materials, but the desired mechanical properties must be considered alongside the other properties of that material.

For food packaging applications and organic electronics, protective coatings with good moisture/oxygen barrier properties are required. The pinhole-free nature of Al_2_O_3_, ZrO_2_, and TiO_2_ ALD coatings—and especially their multilayer combinations—are interesting for these applications [1,2,3,4,5,6]. Thin films of Y_2_O_3_ are considered to be a promising corrosion-resistant and protective coating for deposition chambers and other three-dimensional metal parts in the semiconductor industry [7]; high temperatures are often involved with semiconductor processing, and hence management of temperature-induced stresses is of high importance [8,9,10,11]. Earth-abundant semiconductors ZnO and Al/ZnO are interesting for flexible electronics, where materials have to be mechanically compatible with roll-to-roll processing and flexing in the end applications; flexible thin-film transistors have been demonstrated with ALD-fabricated layers as both the channel and the dielectric materials [12,13]. All these fields share the need for a quantified understanding of mechanical robustness of the coating materials and would benefit from guidelines for smart coating designs via thin-film engineering. The present knowledge on the mechanical straining properties of ALD-fabricated thin films is, however, scarce. Straining behavior of ALD Al_2_O_3_ thin films of has been studied extensively [2,14,15,16,17,18], apart from the effect of the deposition temperature. These properties for ALD-deposited TiO_2_ [17,19] and ZnO [20] have gained less attention.

In this work, we extend the research on the mechanical properties of the ALD-fabricated thin films to a wider range of application-relevant materials, and explore thin-film engineering via multilayering and residual-strain control as means for tailoring the mechanical properties. Mechanical fracture properties are studied via uniaxial tensile testing for Al_2_O_3_, ZnO, TiO_2_, ZrO_2_, and Y_2_O_3_ thin films and their selected multilayer combinations. The deposition temperature (135–220 °C) strongly influences the residual strain, which is shown to dictate the fracture properties for Al_2_O_3_ films, while doping Y_2_O_3_ with Al or nanolaminating it with Al_2_O_3_ are found as effective means for crystal-size engineering of Y_2_O_3_ films, and for tailoring their fracture properties.

## 2. Materials and Methods 

### 2.1. Experimental

The film structures studied in this work are shown in Figure 1. The Al_2_O_3_, ZnO, and Y_2_O_3_ films were first grown directly on both silicon (with native oxide surface) and polyimide substrates. However, for ZnO, Y_2_O_3_, ZrO_2_, and TiO_2_ films, smaller thicknesses were observed on polyimide than on silicon, which we attributed to worse film nucleation on the polyimide substrate. As the use of Al_2_O_3_ as an intermittent layer has proven beneficial for the fabrication of smooth and uniform ALD films on polymer substrates [21,22], an Al_2_O_3_ nucleation layer was implemented for the interface between the polyimide and the material of interest, as shown in Figure 1b. This Al_2_O_3_ layer was deposited in the same ALD run prior to the subsequent deposition of the other oxide materials (no exposure to air or break of vacuum). To control the crystallinity of Y_2_O_3_, two composite formulations were prepared and studied from Al_2_O_3_ and Y_2_O_3_; that is, Al-oxide doping (Figure 1c) and nanolaminating (Figure 1d). Aluminum oxide doping of Y_2_O_3_ was achieved by adding one ALD Al_2_O_3_ cycle every ten ALD Y_2_O_3_ cycles, meaning a doping level of Y_2_O_3_:Al_2_O_3_ of 10:1.

The films were deposited with an ALD reactor (Arradiance, GEMStar XT, Littleton, MA, USA) using the precursors and the temperatures listed in Table 1. All the metal precursors were acquired from Strem Chemicals and H_2_O was used as an oxidizer in all the ALD processes. All the depositions, except that of ZnO, were monitored in situ using a quartz crystal microbalance (Tempe, Colnatec, Greenville, SC, USA). Films were grown on DuPont polyimide strips (Kapton 200HN50 µm thick) and native-oxide covered silicon (100) substrates in the same ALD run. Al_2_O_3_ was deposited at the temperature range of 135–220 °C, while for the other materials, the deposition temperature was 200–220 °C (155 °C for ZnO).

The ALD was carried out as a viscous flow type process for ZrO_2_ and TiO_2_ films, while for the other materials, the precursor pulsing was realized in the so-called exposure (or stop-flow) mode. In the exposure mode, the exhaust valve of the deposition chamber is closed during each precursor pulse and during a desired period of time following the pulse. The unreacted precursors and by-products were purged away using a higher argon gas flow (200 sccm for YMeCp (tris(methylcyclopentadienyl)yttrium) and 100 sccm for the other precursors) than during the precursor pulsing step (30 sccm for the YMeCp process, 40 sccm for the others). For YMeCp, argon was injected (at around 1.4 bar for 20 ms) into the precursor container prior to pulsing in order to facilitate the delivery of the precursor by viscous gas transport. For Y_2_O_3_, the ALD process consisted of 400 cycles of 1500 ms YMeCp pulse with 1 s exposure and 180 s purge followed by 50 ms H_2_O pulse with 2 s exposure and 70 s purge. For Al*_2_*O_3_, the ALD process consisted of 440 cycles of 30 ms TMA (trimethylaluminum) pulse with 1 s exposure and 50 s purge followed by 50 ms H_2_O pulse with 1 s exposure and 60 s purge. For ZnO, the ALD process consisted of 500 cycles of 100 ms DEZn (diethylzinc) pulse with 0.5 s exposure and 60 s purge followed by 50 ms H_2_O pulse with 0.5 s exposure and 60 s purge. For TiO_2_, the ALD process consisted of 1000 cycles of 800 ms pulse of TTIP (titanium(IV) isopropoxide) with 30 s purge followed by two 40 ms H_2_O pulses 1 s apart of each other with 22 s purge. The use of dual pulsing allows better control of the reaction time between the precursors. For ZrO_2_, the ALD process consisted of 340 cycles of 150 ms pulse of ZrBuO (Zirconium(IV) t-butoxide) with 65 s purge and 50 ms pulse H_2_O pulse with 45 s purge.

The thickness and density of the films were obtained using the X-ray reflectivity (XRR, Bruker AXS, D8 Discover, Billerica, MA, USA) technique for the films deposited on silicon substrates. The incident beam (Cu K_α_) was conditioned by a Göbel mirror, a 0.1° divergence slit, and a 0.1° anti-scatter slit. The measurements were done in *θ*–2*θ* geometry and the reflectivity patterns were analyzed by fitting the data to a physical model using DIFFRAC LEPTOS (Bruker) software. The thickness values for the films deposited on the polyimide substrates were obtained by scanning electron microscope (SEM, Hitachi S-4800, Tokyo, Japan) imaging of cross sections prepared by focused Ga-ion beam milling (FIB-SEM: Helios 600, FEI, Hillsboro, OR, USA). Pt-carbon material was locally deposited by Ga-FIB to protect the surface during the FIB cross-sectioning. Cross sections of cracks and buckles in the ALD films after the tensile experiment were performed to investigate crack morphologies and interface failure modes.

The crystallinity of the thin films was investigated by the grazing incidence X-ray diffraction (GIXRD) technique (Bruker AXS, D8 Discover diffractometer, Billerica, MA, USA). The diffractometer was operated at a fixed angle of incidence of 0.5° using Cu K_α_ (*λ* = 1.5418 Å) radiation, scanning the 2*θ* angle in the range of 25°–80°. The incident beam was conditioned with a Göbel mirror and a 0.6 mm slit, while the diffracted beam was passed through an equatorial soller slit, 0.0125 mm Ni filter, and 9.5 mm slit to a 0D detector. The scan was carried out in steps of 0.02°, and the measurement time for each step was 2 s. In addition, to observe possible preferred crystal orientation in the films, symmetric *θ*/2*θ* Bragg–Brentano scans were also performed (with an offset of −1°); for these scans, a 1D detector with an opening of 2.174° was used (data not shown).

X-ray photon spectroscopy (XPS, Phi Quantum 2000, Eden Prairie, MN, USA) was performed for some of the ALD films in a Quantum2000 from Physical Electronics using a monochromatized Al K_α_ source (1486.6 eV). This was done to measure the carbon contamination in the materials and stoichiometry of the films. The Al_2_O_3_ films have been characterized in a previous publication [23].

A displacement controlled tensile stage (MTI 8000-0010, Albany, NY, USA) was used to study the fracture behavior of the films deposited on polyimide substrates. The uniaxial tensile strain experiments were performed at a constant strain rate of 1.7 × 10^−4^ s^−1^. The surface of the samples was monitored in situ with a digital optical microscope (Keyence 500F, Mechelen, Belgium) and a series of images were recorded at strain intervals of 0.015%. Digital image tracking from three pairs of points on the sample surface was used to determine the strain. Polyimide substrate strips for the ALD film growth with a gauge section of 5 × 35 mm^2^ were prepared using a paper cutter. Each ALD film material was deposited on 5–6 polyimide strips (in the same ALD run), and these were subjected to tensile strain measurements and analysis; the results reported hence represent an average over 5–6 measurements for each material.

### 2.2. Theory

The crack density evolution with tensile strain was analyzed using a two-parameter Weibull distribution at small strains. The cohesive strain of the coating is obtained by
(1)εcoh=β(l0/1.5l¯sat)1α·Γ(1+1α)
where ${\bar{l}}_{sat}$ is the mean crack fragment width at saturation (equal to the inverse value of saturated crack density),$\;l_{0}$ is a normalizing factor equal to 1 (µm), Γ is the mathematical gamma function, and  α and β are Weibull strain distribution parameters obtained from fitting the fragment width versus strain graphs. The procedure to obtain the Weibull parameters and the related cohesive strains *ε**_coh_* of the films is explained in detail elsewhere [24,25,26]. The interfacial shear strength, *τ* (Pa), can be obtained from the saturated crack density, *sCD* (m^−1^), at high tensile strains using the Kelly–Tyson approach [27]:(2)$$\tau =1.337\cdot E\cdot h\cdot \varepsilon_{coh}\cdot sCD$$
where *h* (m) and *E* (Pa) are film thickness and elastic modulus, respectively. Owing to the lack of elastic moduli for the ALD films, values are reported as strain instead of strength (dividing Equation (2) by the *E*) in Table 2 for all the studied materials. For Al_2_O_3_ films, strength data are provided in Table 3.

Furthermore, when the residual film stress $\sigma_{r\;}$ is known, the energy release rate G (J/m^2^) can be calculated using the following expression [16,28,29]:(3)$$G=\frac{\pi h}{2E}{\left(\frac{E\varepsilon }{1-\nu^{2}}+\sigma_{r}\right)}^{2}g\left[D_{1},D_{2}\right]$$
where *v* is the film’s Poisson’s ratio and *ε* is the applied tensile strain. The non-dimensional integral of crack opening displacement *g* [29] is a non-linear function depending on the elastic mismatch between the film and the polymer substrate described by Dundurs’ parameters *D_1_* and *D_2_* [28]. When the energy release rate *G* in Equation (3) reaches the critical value *G_C_* to fracture the films, that is, when the applied tensile strain equals the strain at crack onset, $\varepsilon_{COS}$, it is related to the film’s tensile (mode I) fracture toughness, $K_{IC}$ (Pa·m^0.5^),
(4)$$G_{C}=K_{IC}^{2}/E$$

The residual film stress is composed of a growth-induced intrinsic stress $\sigma_{i\;}$ and the thermal stress $\sigma_{T\;}$ owing to the thermal mismatch between the film and substrate, $\sigma_{r\;}=\sigma_{i\;}+\sigma_{T\;}$. Residual tensile film stresses have been reported for ALD Al_2_O_3_ films on Si [30]. For calculating residual film stresses on polyimide—which shrinks considerably with respect to Al_2_O_3_ ALD films upon cooling to room temperature after ALD—the expression
(5)$$\sigma_{T\;}=\frac{E}{\left(1-v\right)}\left(\alpha_{s}-\alpha_{f}\right)∆T$$
was used, where *α* is the coefficient of thermal expansion (ppm K^−1^) of the polyimide substrate (subscript s) and film (subscript f), respectively, and Δ*T* is the temperature difference (K).

The ability of the thin ALD films to adapt to changes of shape can be evaluated in terms of the critical bending radius, at which cracking of the film would occur on the flexible substrate upon its bending [31],
*R* = (*h* + *h_s_*)/(2*ε_cos_*)(6)

Of note is that the bending radius is dependent on the polymer substrate thickness, *h_s_*; hence, it is a materials system property and not the property of the ALD film alone.

## 3. Results

### 3.1. Crystallinity and Composition of the Films

The crystal structure of the thin-film materials was studied by the GIXRD technique for the films deposited on Si substrates in order to determine potential correlations between film crystallinity and mechanical properties. The distinct diffraction maxima seen in the GIXRD patterns for the ZnO, Y_2_O_3_, Y_2_O_3_ on Al_2_O_3_, and ZrO_2_ on Al_2_O_3_ samples indicate polycrystalline structures for these thin films (Figure 2). As the TMA/H_2_O ALD process is known to yield amorphous Al_2_O_3_ material (at least within the TEM observation limit of 0.9 nm) [32], Al_2_O_3_ layers do not contribute to the diffraction patterns of the nanolaminate and the bilayer samples. The diffraction maxima for the ZnO film were indexed to the hexagonal wurtzite structure, while the Y_2_O_3_ films were found to crystallize in the cubic crystal structure, as has been observed in the literature for the used ALD process [33].

The GIXRD patterns for the Y_2_O_3_/Al_2_O_3_ nanolaminate and Al-doped Y_2_O_3_ were characteristic for (nearly) amorphous materials; the low-intensity broad features seen at around 32° (with possible contributions from the 222 and 400 maxima) might indicate the presence of very small nanocrystals. Therefore, nanolaminating and doping were found to be efficient tools to tailor the crystal size of the Y_2_O_3_ films, confirming the previous observations in the literature [34]. The marked decrease in the crystal size is likely because of the AlO*_x_* layers disrupting the growth of Y_2_O_3_ nanocrystals, similar to the effect of TMA pulsing in between DEZn/H_2_O cycles for the deposition of ZnO films [32,35,36,37].

The absence of the diffraction peaks for the TiO_2_ films indicated an amorphous character for this material, which is in line with the deposition temperature of 200 °C, being around the typical limit for the amorphous-to-crystalline transition for the TTIP/H_2_O process [38].

The diffraction pattern for the ZrO_2_ films showed diffraction peaks labeled as (i–iii). The position of the peaks (i–iii) could be ascribed to the monoclinic crystal structure of ZrO_2_ [39,40,41], with (i), (ii), and (iii) corresponding to the 111, 200, and −122 peaks, respectively. However, the broadness of the peaks leaves room for ambiguity, as the tetragonal structure of ZrO_2_ also shows peaks at close proximity [40,41]. Kukli et al. observed a tetragonal structure for thin ZrO_2_ layers in a nanolaminate structure, while a full-thickness film had a monoclinic structure [40]. Hence, it could be that our films have a mixed monoclinic–tetragonal structure, with the film growth possibly beginning with tetragonal structure and continuing with monoclinic structure after a certain thickness. In addition, we performed XRD scans in the symmetric Bragg–Brentano mode (data not shown), which indicated that the ZnO and Y_2_O_3_ films had 002 and 222 preferred orientation, respectively.

The chemical composition of the films was studied with XPS. The chemical composition of Al_2_O_3_ films can reach stoichiometry and low levels of hydrogen contamination (< 4% from unreacted –OH groups) at 220 °C. Consequently, the mass density increases to 3.3 g/cm^3^, with a carbon content below 1% (from unreacted ligands), as reported in our previous study [23]. The XPS measurements show that the carbon contamination of TiO_2_ films is about 1.5% and stoichiometric at 200 °C. The ZnO films are stoichiometric at 155 °C with a carbon contamination below the detection limits of XPS. ALD of Y_2_O_3_ films at 220 °C using YMeCp had a composition of 54% O, 43% Y, and 3% C, see the depth profile in Appendix A. Al_2_O_3_ films deposited at 220 °C were similarly measured to have composition of 35% Al and 65% O. The XPS depth profile of the Y_2_O_3_/Al_2_O_3_ nanolaminate is shown in Appendix A.

### 3.2. Fragmentation of the Films

The fragmentation process was qualitatively similar for all the samples studied in this work. As an example, optical micrographs of the surface of an Y_2_O_3_ film under varying tensile strain are shown in Figure 3. The non-tensioned sample in Figure 3a shows the surface roughness and inhomogeneity of the polyimide surface visualized through the optically transparent Y_2_O_3_ film. The surface inhomogeneities of the polyimide did not affect the crack evolution. Figure 3b shows the sample at 0.7% tensile strain, where the first channel cracks appeared. At strain values slightly above the crack onset strain, the crack density increased rapidly. Under tensile strain of 4.6%, about half of the saturation crack density had already been reached (Figure 3c), and the saturation was finally reached around strain values of 9%–10%.

Owing to contraction of the substrate perpendicular to the tensile direction, transverse buckling/cracking occurred at strains of 9%–10% [42]. When the strain was further increased, more transverse buckles were formed, as shown in Figure 3d. The SEM analysis revealed the transverse buckles formed during testing had usually cracked at their peak, their base, or both; images are shown in Figure 4 for ZnO and Al_2_O_3_. Of note is that the small size of the buckles indicates a relatively good adhesion of the films to the polyimide substrate.

Figure 5 shows SEM observations of the FIB cross-section cuts of the channel cracks formed during tensile stretching of the films. We observed that only the thin ALD oxide films had cracked. Propagation of the cracks to the polyimide substrate was not observed. Additionally, we did not observe signs of the ALD precursors diffusing into the polymer and forming sub-surface growth material. Both observations can be attributed to the use of the present polyimide as a flexible carrier substrate, instead of other materials. For instance, when using polyethylene-terephthalate (PET) as a carrier substrate for ALD Al_2_O_3_ films, sub-surface growth and substrate cracking have been observed [43].

### 3.3. Mechanical Properties of the Films

Figure 6 shows the quantitative evolution of the crack density versus uniaxial tensile strain for all the film materials containing Y_2_O_3_. The interesting observation is that the polycrystalline Y_2_O_3_ directly grown on polyimide had the highest saturated crack density value (Figure 6a). The introduction of the 20 nm thick interfacial Al_2_O_3_ layer reduced the crack density by a factor of 1.4 (Figure 6b). On the basis of this behavior, we decided to further engineer the film composition and microstructure, by tailoring the nanocrystal size of Y_2_O_3_ through doping with one Al_2_O_3_ cycle after every ten Y_2_O_3_ cycles (Figure 6c), and through the fabrication of the Y_2_O_3_/Al_2_O_3_ nanolaminate structure (Figure 6d). The saturated crack density decreased by a factor of around two in comparison with the Y_2_O_3_ films.

Interestingly, the Y_2_O_3_/Al_2_O_3_ nanolaminate showed an additional plateau in the crack density versus tensile strain curve. We speculate that the nanolaminate structure introduces an additional fragmentation mechanism at higher strains. This remains to be studied in detail, however, being outside the scope of this work. The saturated crack density determined here for the polycrystalline ZnO films on polyimide is a factor of two higher than what has been determined previously for polycrystalline ZnO/Al_2_O_3_ nanolaminates on PET with an interfacial Al_2_O_3_ layer (Table 2) [43].

The crack onset strain and the saturated crack density values are compiled in Table 2. Both quantities have been observed to scale with the film thickness by the $h^{-1/2}$ relationship [26,44]. In order to allow for a thickness-independent comparison of the crack onset strain and the saturated crack density values, we normalized these quantities to 50 nm total film thickness (Table 2 and Figure 7). In Figure 7, the position on the normalized crack onset strain axis is determined not only by the strength of the film material, but also by the internal residual film strain developing during ALD, and by the thermal mismatch upon cooling to room temperature. Hence, the data are specific to the film-substrate material system, not only to the film material.

Critical fracture onset strain for different thicknesses of Al_2_O_3_ films on PEN deposited at 155 °C have been reported to be 1.18%, 0.95%, and 0.52% for 25, 40, and 80 nm thick Al_2_O_3_ films, respectively [14]. Thus, these results agree well with our observations. Leterrier et. al have reviewed the work done on PVD films, and reported far better crack onset strain (COS) values for 400 nm thick SiO_2_ on polyimide 1.07% and for similar SiN 0.97% [45].

The saturated crack density depends on the parameters in Equation 2; that is, the interfacial shear strength, the fracture strain, and the Young’s modulus of the film material. Interestingly, the polycrystalline films of Y_2_O_3_ and ZnO (as measured by GIXRD; Figure 2) group in the zone with the highest saturated crack density and the lowest crack onset strain. This could be explained by larger tensile residual stress forming in these nanocrystalline films during film growth in comparison with the stress forming during the growth of amorphous oxide films. An exception is the polycrystalline ZrO_2_ on Al_2_O_3_ film. Interestingly, the ZrO_2_ and TiO_2_ films on interfacial Al_2_O_3_ layers show the highest crack onset strain values, together with reasonably low crack density values. Another feature of note in Figure 7 is the role of the amorphous interfacial Al_2_O_3_ layer. The interfacial layer reduces the saturated crack density and increases the crack onset strain with respect to Y_2_O_3_ directly grown on polyimide. Once the interfacial Al_2_O_3_ layer is introduced, it does not seem to matter if the material deposited on top of it is polycrystalline or amorphous, as the (X-ray) amorphous aluminum-doped Y_2_O_3_ and polycrystalline Y_2_O_3_ show very similar characteristics in Figure 7. The Y_2_O_3_/Al_2_O_3_ nanolaminate (high Al_2_O_3_ content) on the interfacial Al_2_O_3_ layer has lower saturated crack density and slightly higher crack onset strain than the Al-doped Y_2_O_3_ (low AlO*_x_* content), and these values seem to approach those for the Al_2_O_3_ film deposited at 220 °C. Therefore, the properties for the Y_2_O_3_-Al_2_O_3_ mixtures seem to be governed by the composition ("rule of mixtures") of the multilayer films, rather than the crystallinity of the Y_2_O_3_ layers. The data for the Y_2_O_3_-Al_2_O_3_ mixtures show how the fragmentation properties of thin films can be tuned via multilayer engineering, where residual strain differences between the different layers likely play a key role.

Residual-strain engineering was studied in detail for the Al_2_O_3_ films. The amorphous Al_2_O_3_ films deposited at different temperatures did not show a large difference for the normalized saturated crack density values (Figure 7). However, a substantial difference was obtained for the normalized crack onset strain values. This trend is probably linked to the temperature dependence of the hydrogen content (9 at.%, 7 at.%, and 4 at.% for 135, 155, and 220 °C, respectively) [23] and mass density [23], properties that likely contribute to the intrinsic stresses in the films. Intrinsic stresses for ALD-fabricated Al_2_O_3_ films have been reported by Ylivaara et al. [30], which, together with the thermal expansion coefficients for the Al_2_O_3_ films ($\alpha_{f}=4.2\;ppm/K$) [46] and for polyimide substrate ($\alpha_{s}=21.8,\;\;23.3,\;\mathrm{and}\;27\;ppm/K$ for temperatures of 135, 155, and 220 °C, respectively) [47], enable the calculation of residual strain values for the Al_2_O_3_ films on polyimide via Equation (5). The data by Ylivaara are included in Table 3, and it can be seen that the Al_2_O_3_ films transit from tensile to increasingly compressive residual strain with the increasing ALD temperature. Compressive/tensile residual strain acts as a negative/positive strain offset in tensile-strain experiments. The “intrinsic” crack onset strain values obtained here (corrected for residual strain as $\varepsilon_{COS\;Al2O3}=\varepsilon_{COS\;on\;PI}+\varepsilon_{r\;on\;PI}$) are 0.89% ± 0.05%, independent of the ALD growth temperature. Accordingly, most of the difference between the crack onset strain values for the Al_2_O_3_ samples (Table 2 and Figure 7) can be explained by the tensile intrinsic strain in the Al_2_O_3_ films and by the compressive strain induced by the post-deposition cooling of the polyimide substrate. This finding highlights that control over the residual strain (deposition temperature) provides us with a tool for tailoring the fracture properties of thin-film coatings.

The cohesive strain values for the film-substrate material systems follow the trend of the crack onset strain values, that is, high crack onset strain values correspond to high cohesive strain values (Table 2). Nevertheless, the material-to-material variation was less for cohesive strain than for crack onset strain. Cohesive strain takes into account the kinetics of the fracturing, that is, how fast the crack density increases with the increasing strain after the initial fracture. This may be subject to random film defects depending on the ALD growth, crystallinity of the film, or contamination content. The Al_2_O_3_/Y_2_O_3_ nanolaminate, the polycrystalline Y_2_O_3_, and the Al_2_O_3_ film deposited at 135 °C have the highest cohesive strain with respect to their crack onset strain (about a factor of 1.6 higher), meaning that the crack density increased more slowly with the increasing strain for these samples. Polycrystalline ZnO has the lowest cohesive strain with respect to crack onset strain (about a factor of 1.2 higher), which indicates that the cracking continues very quickly with the increasing strain once it has begun. The factors for the other studied materials, all with interfacial Al_2_O_3_ layers, fall in the range of 1.2–1.6.

Cohesive film strain and the interfacial shear strain are listed in Table 2. Both of these strains involve the crack onset strain in their calculation, which in turn depends on the residual film and thermal mismatch strains. Consequently, the cohesive film strains and interfacial shear strains in Table 2 relate to the film-substrate material systems. In Figure 8, we plot the interfacial shear strain versus the dimensionless term 1.337 × ***sCD*** × ***h*** in the interfacial shear strain equation (Equation (2)), which allows us to situate film materials according to their cohesive strains (which are naturally presented as straight lines in this plot). Interestingly, the film materials distinctively group along two straight lines that correspond to cohesive strain values of 1.2% and 1.5%. The 1.5% line is by tendency populated by the amorphous oxides (with the exception of ZrO_2_) deposited on the amorphous interfacial Al_2_O_3_ layer. However, no clear conclusion can be drawn on the role of the amorphous interfacial Al_2_O_3_ layer on improvement of adhesion, as the 1.2% line is also partly populated by samples with the interfacial Al_2_O_3_ layer. Polycrystalline samples have a tendency to group along the 1.2% line. Moreover, for the ALD Al_2_O_3_ films, the cohesive film strength and interfacial shear strength were obtained by multiplying the corresponding strain values in Table 2 by the elastic modulus. The elastic modulus values for the ALD Al_2_O_3_ thin films are available from the literature [30,32,46,48,49,50,51,52] and the value of 150 GPa was used for the sample deposited at 135 °C, while 170 GPa was used for the samples deposited at 155 °C and 220 °C (Table 3).

The adhesion between the film materials and the polyimide substrate was qualitatively good for all the studied materials, as delamination of the ALD films from the substrate was not observed during the tensile testing. The interfacial shear strain and strength give quantified insight into the adhesive properties of the film–polymer interface. The obtained interfacial shear strain values of 0.03% to 0.04% lie in the range of what has been reported for inorganic films on polymers. Higher values for the interfacial shear strains of 0.09%–0.14% have been reported for 10–110 nm thick PECVD SiO*_x_* films on polyamide-12 and polyethylene-terephtalate (PET); these values correspond to interfacial shear strengths of 74–113 MPa [53]. For ZnO, Al_2_O_3_, and their nanolaminate ALD films on PET, the interfacial shear stresses of 59 MPa (strain 0.04%), 15 MPa (strain 0.01%), and 16 MPa to 24 MPa (strain 0.01% to 0.02%) were reported [43]. For 70 nm thick sputtered indium-zinc-oxide films on polyethylene-napthalate, an interfacial shear strength of 16 MPa (strain of 0.014%) has been reported [54]. Moreover, any significant difference in the adhesion of Al_2_O_3_ on polyimide was not observed within the studied temperature range.

The calculation of the tensile fracture toughness according to Equations (3) and (4) allows for the comparison of thin-film materials with bulk samples. There is no clear trend with regard to the deposition temperature. Our values that fall in the range of 1.3–1.7 MPa·m^0.5^ are similar to the previously reported values of 1.5–2.2 MPa·m^0.5^ for ALD Al_2_O_3_ [16]. Moreover, this value is lower than the typical values for bulk amorphous Al_2_O_3_ (2.5–5 MPa·m^0.5^) [55]. This could be because of the naturally incorporated hydrogen in ALD Al_2_O_3_ thin films deposited via the TMA/H_2_O process [23].

Moreover, flexibility of the thin films was evaluated in terms of the critical bending radius from Equation (6). The values for the critical bending radius are low, and fall in the range of 2–3.6 mm for all studied films. Therefore, the chemical composition of the metal-oxide films (in the present thickness range) does not play a big role for the practical applicability of these thin-film materials for applications where bending is required; the films are suitable to withstand roll-to-roll processing and to conform to substantially curving surfaces.

## 4. Conclusions

We have presented a systematic study of mechanical fracture properties for some of the most common atomic layer deposited (ALD) metal-oxide thin films. Films of 50–70 nm thickness were deposited on flexible polyimide (Kapton) carrier substrates and characterized via uniaxial tensile experiments under strain values up to 20%. FIB cross sections did not show any SEM-visible sub-surface growth of the ALD materials in the polyimide substrate. Use of an interfacial/adhesion Al_2_O_3_ layer improved the nucleation of some of the oxide materials on the polyimide substrate. Delamination of the films from the polyimide substrate was not observed for any of the film materials and the values for interfacial shear strain ranged from 0.027% to 0.05% for all materials. Crack onset strain values (normalized to 50 nm thickness) varied from 0.7% to 1.3% for the materials investigated, with TiO_2_ and ZrO_2_ on interfacial Al_2_O_3_ showing the largest crack onset strain values. The cohesive strain values ranged between 1.2% and 1.5%.

Thin-film engineering through the use of an interfacial Al_2_O_3_ layer, nanolamination with Al_2_O_3_, or doping with Al were shown as effective means for tailoring the fracture properties of the Y_2_O_3_-based thin films; a nearly factor-of-two decrease for the saturated crack density and a nearly factor-of-two increase for the crack onset strain were obtained via the nanolamination approach. Thin-film engineering through residual-strain control was studied for the Al_2_O_3_ films; the fact that the residual strain becomes increasingly compressive with increasing deposition temperature was exploited, and it was shown that the crack onset strain and the cohesive strength/strain values can be substantially increased simply by increasing the deposition temperature. The ALD growth temperature and thin-film multilayer engineering are thus highlighted as important tools for designing mechanically resistant stretchable/flexible coatings.

## Figures and Tables

**Figure 1 nanomaterials-10-00558-f001:**
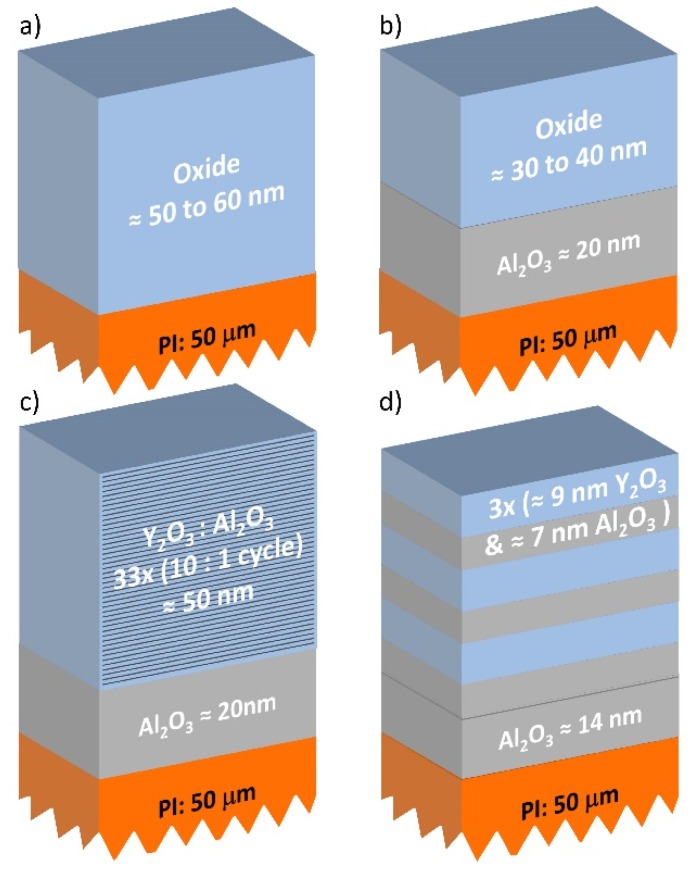
Schematic representation of the thin film geometries studied in this work: (**a**) oxide on polyimide (Al_2_O_3_, ZnO, Y_2_O_3_), (**b**) oxide on interfacial Al_2_O_3_ on polyimide (Y_2_O_3_, TiO_2_, ZrO_2_), (**c**) aluminum oxide-doped Y_2_O_3_ on interfacial Al_2_O_3_ on polyimide, (**d**) Al_2_O_3_–Y_2_O_3_ nanolaminate on interfacial Al_2_O_3_ on polyimide.

**Figure 2 nanomaterials-10-00558-f002:**
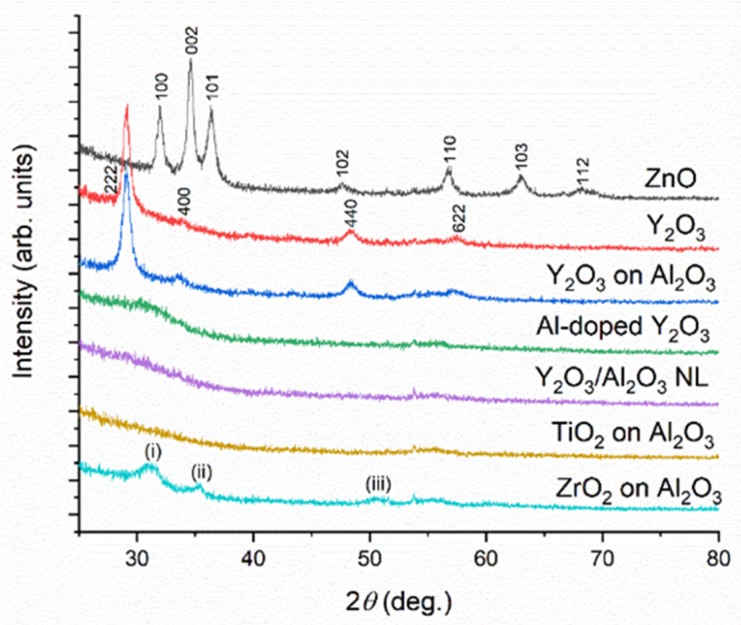
Grazing incidence X-ray diffraction (GIXRD) patterns of the film materials grown on silicon.

**Figure 3 nanomaterials-10-00558-f003:**
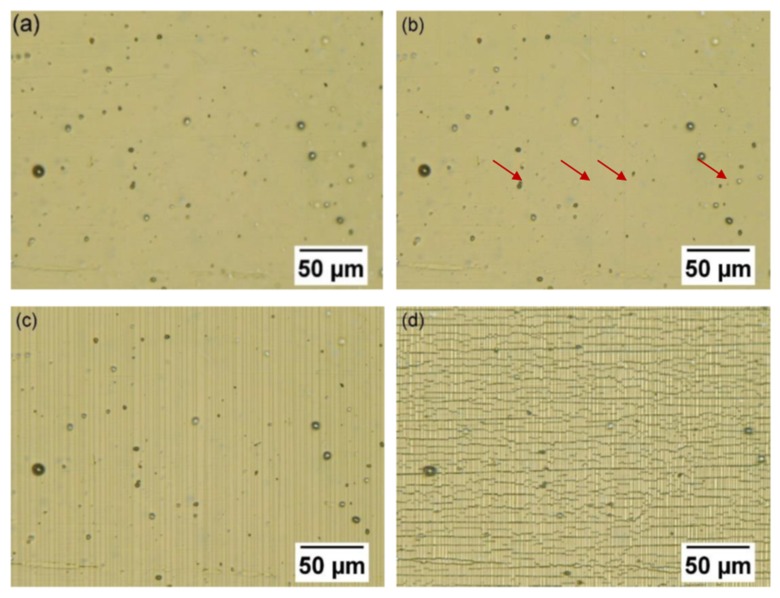
Optical microscope images of the fragmentation process of the 46 nm thick Y_2_O_3_ film grown directly on the polyimide substrate. The strain was applied horizontally in the field of view of the image. (**a**) Unstrained Y_2_O_3_ specimen. (**b**) At 0.7% tensile strain, the first cracks were observed. (**c**) At 4.6% tensile strain. (**d**) At 20.3% tensile strain, many transverse buckles were formed.

**Figure 4 nanomaterials-10-00558-f004:**
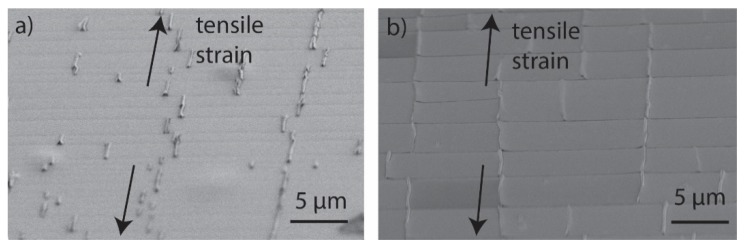
Scanning electron microscope (SEM) overview of the fractured samples after tensile straining up to 12% showing the channel cracks and the transverse buckles/crack owing to lateral contraction. (**a**) ZnO on polyimide from a 55° angle and (**b**) Al_2_O_3_ deposited at 155 °C on polyimide from a 52° angle.

**Figure 5 nanomaterials-10-00558-f005:**
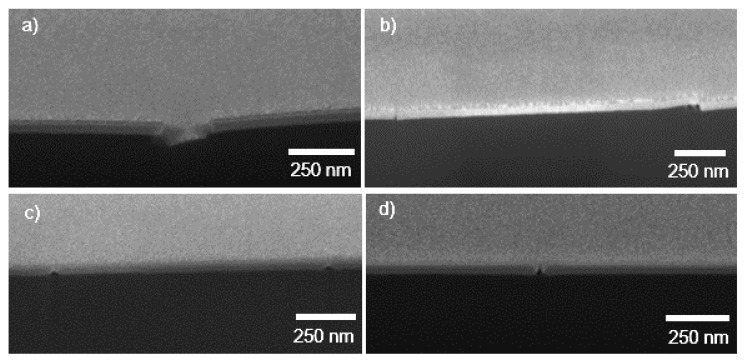
SEM images of focused Ga-ion beam (FIB) cut cross-sections (52° angle) of atomic layer deposition (ALD) films on polyimide. (**a**) Al_2_O_3_ 220 °C, (**b**) ZnO, (**c**) ZrO_2_ on Al_2_O_3_, and (**d**) Y_2_O_3_ samples. No sub-surface growth into the polyimide was observed.

**Figure 6 nanomaterials-10-00558-f006:**
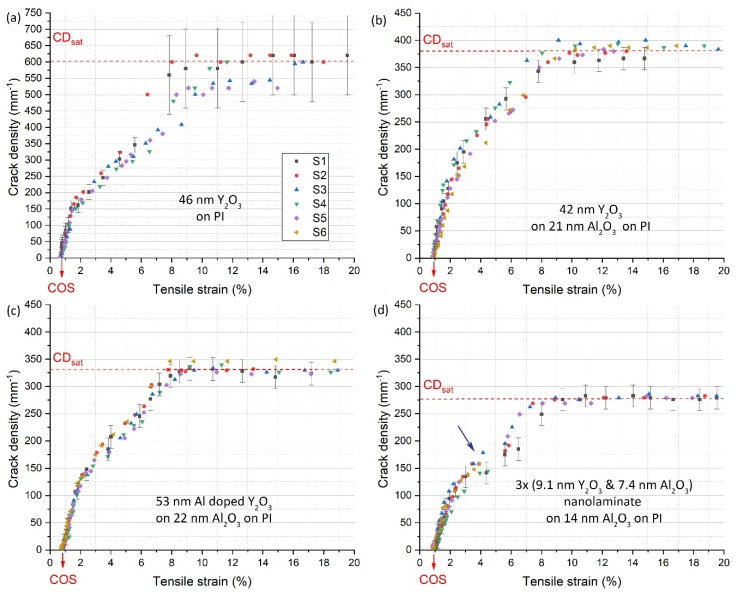
Crack density versus tensile strain plots for the Y_2_O_3_-based samples. (**a**–**d**) The specific film structure, the saturated crack density (CD_sat_), and the crack onset strain (COS) are indicated. (**d**) The arrow indicates the formation of an additional distinct crack-density plateau. Uncertainties take account for sample variations, fitting residuals, and measurement accuracies. For clarity, the error bars are shown only for one sample per film material.

**Figure 7 nanomaterials-10-00558-f007:**
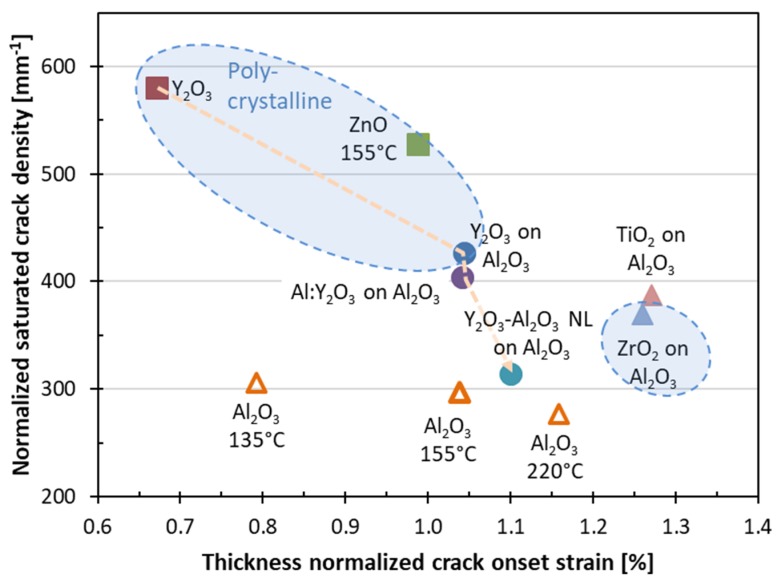
The crack density values vs. crack onset strain values, as normalized to 50 nm thickness. Y_2_O_3_-based films are connected by the yellow dashed line to indicate improvements owing to the interfacial Al_2_O_3_ film, Al-doping, and nanolaminating. The blue dashed lines enclose the data for the polycrystalline films. The errors bars (omitted for clarity) are around 10% for the saturated crack density values and 10%–15% for the crack onset strain values.

**Figure 8 nanomaterials-10-00558-f008:**
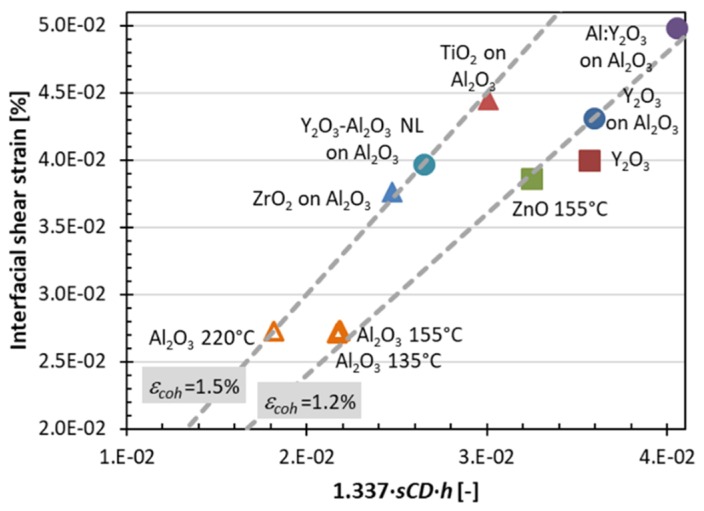
Interfacial shear strains versus the saturated critical crack density (*sCD*) times total film thickness (*h*) for all films on polyimide. Data points along a straight line feature the same cohesive film strain. The error bars were omitted in the graph for clarity and are around 10% to 30% for the interfacial shear strain and 20% for the x-axis; the exact values are listed in Table 2.

**Table 1 nanomaterials-10-00558-t001:** Atomic layer deposition (ALD) metal oxide film materials and metal precursors used together with H_2_O to deposit the films. Thickness and density of the films grown on Si were obtained by X-ray reflectivity.

Sample	Metal Precursor ^(1)^ and Temperature	ALD (°C)	Number of ALD Cycles	Total Film Thickness (nm) on Si	Thickness Top Oxide (nm)	Interfacial Al_2_O_3_ (nm)	Density (g cm^−3^)
Al_2_O_3_	TMA 25 °C	135	440	53	-	-	3.00
Al_2_O_3_	TMA 25 °C	155	440	55	-	-	3.2
Al_2_O_3_	TMA 25 °C	220	440	49	-	-	3.3 [23]
ZnO	DEZn 25 °C	155	500	46	-	-	5.60
Y_2_O_3_	YMeCp 150 °C	220	400	55	-	-	4.50
Y_2_O_3_ on Al_2_O_3_	YMeCp 150 °C	220	190 + 280	63	42	21	^(2)^
Al-doped Y_2_O_3_ on Al_2_O_3_	YMeCp 150 °C	220	195 + 33 × (1:10)	75	53	22	^(2)^
TiO_2_ on Al_2_O_3_	TTIP 75 °C	200	190 + 1000	58	37	21	^(2)^
ZrO_2_ on Al_2_O_3_	ZrBuO 75 °C	220	190 + 340	50	32	22	^(2)^
3 × NL Y_2_O_3_/Al_2_O_3_ on Al_2_O_3_	YMeCp 150 °C and TMA 25 °C	220	115 + (75/57) × 3	63	3 × (9.1/7.4)	14	^(2)^

^(1)^ The precursor acronyms are TMA = trimethylaluminum, YMeCp = tris(methylcyclopentadienyl)yttrium, DEZn = diethylzinc, ZrBuO = Zirconium(IV) t-butoxide, and TTIP = titanium(IV) isopropoxide. ^(2)^ The density was used as an input parameter for multilayer film samples.

**Table 2 nanomaterials-10-00558-t002:** Crack onset strain, saturation crack density, critical bending radius, cohesive strain, and interfacial shear strain for the studied samples. Thickness-normalized crack onset strains (COS 50 nm) and saturated crack densities (sCD 50 nm) relate to the total film thickness including the interfacial Al_2_O_3_ layer when applicable. Uncertainties include sample variations, fitting residuals, and measurement accuracies. The data represent an average over 5–6 tensile measurements for each material.

Sample	Total Thickness (nm)	Crack Onset Strain (%)	Normalized COS (%) (50 nm)	Saturation Crack Density (mm^−1^)	Normalized sCD (mm^−1^) (50 nm)	Bend Radii (mm)	Cohesive Strain (%)	Interfacial Shear Strain (‰)
Al_2_O_3_ 135 °C	53	0.77 ± 0.1	0.79	300 ± 28	310	3.2	1.3 ± 0.3	0.26 ± 0.07
Al_2_O_3_ 155 °C	55	0.99 ± 0.12	1.0	280 ± 30	300	2.5	1.3 ± 0.1	0.25 ± 0.04
Al_2_O_3_ 220 °C	49	1.2 ± 0.16	1.2	280 ± 28	280	2.1	1.5 ± 0.23	0.28 ± 0.06
ZnO	46	1.0 ± 0.12	0.98	550 ± 35	530	2.4	1.2 ± 0.12	0.4 ± 0.06
Y_2_O_3_	46	0.70 ± 0.09	0.67	610 ± 90	580	3.6	1.1 ± 0.26	0.42 ± 0.12
Y_2_O_3_ on Al_2_O_3_	63	0.93 ± 0.12	1.0	380 ± 36	430	2.7	1.2 ± 0.25	0.38 ± 0.1
Al/Y_2_O_3_ on Al_2_O_3_	75	0.85 ± 0.12	1.0	330 ± 31	400	3.0	1.2 ± 0.2	0.41 ± 0.09
TiO_2_ on Al_2_O_3_	58	1.2 ± 0.15	1.3	360 ± 25	390	2.1	1.5 ± 0.3	0.41 ± 0.09
ZrO_2_ on Al_2_O_3_	50	1.3 ± 0.15	1.3	370 ± 28	370	2.0	1.6 ± 0.2	0.38 ± 0.07
Y_2_O_3_–Al_2_O_3_ NL on Al_2_O_3_	63	0.93 ± 0.11	1.0	280 ± 24	310	2.5	1.5 ± 0.2	0.41 ± 0.07

**Table 3 nanomaterials-10-00558-t003:** Summary of fracture data of Al_2_O_3_ ALD films and their residual strains $\varepsilon_{r}=\varepsilon_{i}+\varepsilon_{T}$ on polyimide. Young’s modulus E and the film’s intrinsic strains $\varepsilon_{i}$ were deduced from Ylivaara [30]. Thermal strains $\varepsilon_{T}$ were calculated with Equation (5). Strength data were calculated from related strain in Table 2. The tensile fracture energy and toughness were calculated according to Equations (3) and (4). Positive signs indicate tensile stress and strain, while negative signs indicate compressive stress and strain.

Al_2_O_3_ ALD (°C)	E (GPa)	$\varepsilon_{i}$ (%)	$\varepsilon_{T}$(%)	$\varepsilon_{r}$(%)	Cohesive Strength (GPa)	Interfacial Shear Strength τ (MPa)	Fracture Energy $G_{C}$ (J m^−2^)	Fracture Toughness $K_{IC}$ (MPa·m^0.5^)
135	150	0.27	−0.19	0.07	2.1 ± 0.5	45 ± 11	12 ± 6	1.3 ± 0.7
155	170	0.19	−0.25	−0.05	2.2 ± 0.2	43 ± 4	17 ± 8	1.7 ± 0.9
220	170	0.16	−0.44	−0.29	2.6 ± 0.4	48 ± 7	12 ± 6	1.4 ± 0.7

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
