# Peer review of "Thin-Film Engineering of Mechanical Fragmentation Properties of Atomic-Layer-Deposited Metal Oxides"

_nanomaterials, 2020, doi:10.3390/nano10030558_

Round 1
Reviewer 1 Report
Equations and mathematical terms need to be clarified. The characters are blurred.
Line 217 “Moreover” should be replaced by “In addition”
Lines 224-226 The XPS data could be usefully included as supplementary data.
Line 273 Reference missing.
Line 294 “account sample” should be “account of sample”.
Line 298 Format reference correctly.
Line 369 “slower” should be “more slowly”
Line 373 Reference missing.
Line 396 Reference missing.
Reviewer 2 Report
The current paper contains reports on mechanical properties of an impressive compilation of traditional ALD-type of materials. This is a topic that is relatively underinvestivgated in the community. As such, it will gather interest in a selection of the community considering applications.
The science appears well performed and reported. My main criticism with the current paper is in its readability for someone that is not directly in the core of the field presented. I do believe the work will gain readers and interest if the authors invest more time on editing the document and its structure.
Some minor comments:
- The paper contains rather many inconsistencies related to punctuation. This also appears in the figures. The first impression of the paper would clearly have benefitted from an additional read-through before submission.
- Update the numbers in the tables to reflect on use of significant digits.
- Several references has returned in errors in the paper.
Some suggestions:
- The language used in the abstract has a notably more difficult sentence structure than the rest of the paper, and is rather difficult to read if not already having read the whole document. Suggest to rewrite the abstract to a level that provides information for the average reader.
P1L41: “…e.g. due to large temperature gradients”. Do you really mean temperature gradients or should it be temperature variations/fluctuations?
P2L57: “Thin films of …” This sentence must be rewritten since it can be interpreted in many ways. Suggest to separate Al2O3 from ZnO. Also, it is unclear what types of properties you refer to with “…have been studied in detail…”
P2L69: The beginning of this paragraph needs rewriting. At this point in the text, it is unclear to the reader why this selection of materials was chose, and why the types of laminates where produced. The reason behind the selection is introduced in P9L277. Suggest to bring the ideology behind the samples to a very early stage. This should also be mentioned in the abstract, since at the current state, it seems rather arbitrary.
P3L108: The term “ZrBuO” is introduced here, but only explained in the caption of Table 1. This also includes other abbreviations. All these abbreviations must be explained in the text before used.
- Suggest to also include number of cycles used for the different deposition systems in Table 1.
P4L134: “To observe… …(with an offset of -1 degree)…” What does the offset refer to? Why used? To me it appears as something remaining from GIXRD, while this text refers to symmetric mode. There is something I must have misunderstood.
On overall, the authors have spent rather much text on describing, comparing and referring to effects that appear as minor for me as a reader. One example is the crystallization temperature for growth of TiO2. I suggest to condense the text more to focus on the core, i.e. mechanical properties, and rather report on their own observations of things that are not essential to explain. Otherwise, the text is relatively heavy to read.
The compilation of the current set of data is well described within this paper, however, I do lack the relation to what you would expect as results when compared to other techniques of deposition of materials, or other similar. Since this is a relatively understudied area, it would be very nice with a section that puts the current set of data in perspective with the results of alternative synthesis approaches, and alternative combination of materials.
The authors have gone a long way in showing the scattering of data within the current set of experiments. However, since most, if not all, samples within a material system was deposited in one run, I would also like some comments on expected variations between different runs. How much variation have they observed in practice within a material type when multiple runs are used.
